# Joint Modeling of Severe Dust Storm Events in Arid and Hyper Arid Regions Based on Copula Theory: A Case Study in the Yazd Province, Iran

**Tayyebeh Mesbahzadeh [1], Maryam Mirakbari [1], Mohsen Mohseni Saravi [1], Farshad Soleimani Sardoo [1,2] and Nir Y. Krakauer [3,*]** 

[1] Department of Reclamation of Arid and Mountain Regions, Faculty of Natural Resources, University of Tehran, Tehran 1417414418, Iran; tmesbah@ut.ac.ir (T.M.); maryammirakbari@ut.ac.ir (M.M.); msaravi@ut.ac.ir (M.M.S.); farshad.soleimani@ut.ac.ir (F.S.S.)

[2] Department of Natural Engineering, Faculty of Natural Resources, University of Jiroft, Jiroft 7867161167, Iran

[3] Department of Civil Engineering, the City College of New York, New York, NY 10031, USA

\* Correspondence: nkrakauer@ccny.cuny.edu

**Abstract:** Natural disasters such as dust storms are random phenomena created by complicated mechanisms involving many parameters. In this study, we used copula theory for bivariate modeling of dust storms. Copula theory is a suitable method for multivariate modeling of natural disasters. We identified 40 severe dust storms, as defined by the World Meteorological Organization, during 1982–2017 in Yazd province, central Iran. We used parameters at two spatial vertical levels (near-surface and upper atmosphere) that included surface maximum wind speed, and geopotential height and vertical velocity at 500, 850, and 1000 hPa. We compared two bivariate models based on the pairs of maximum wind speed–geopotential height and maximum wind speed–vertical velocity. We determined the bivariate return period using Student t and Gaussian copulas, which were considered as the most suitable functions for these variables. The results obtained for maximum wind speed–geopotential height indicated that the maximum return period was consistent with the observed frequency of severe dust storms. The bivariate modeling of dust storms based on maximum wind speed and geopotential height better described the conditions of severe dust storms than modeling based on maximum wind speed and vertical velocity. The finding of this study can be useful to improve risk management and mitigate the impacts of severe dust storms.

**Keywords:** copula theory; bivariate return period; joint probability; dust storms; maximum wind speed; geopotential height

## 1. Introduction

Evidence suggests that the rate of natural disasters has increased worldwide from 1993 to 2002. Of 2654 natural disasters that occurred during this time period, floods and dust storms were the most common types, accounting for 70% of the total, with drought, landslides, fires, etc., making up the remainder [1].

Dust storms, the result of particle transfer through the air, are an environmental problem and serious natural disaster in arid and semiarid regions. In general, dust storms are caused by wind turbulence, severe winds, and rapid ascents of dust to the upper atmosphere, which requires an increase in vertical and horizontal velocity over the area. The climate and environmental conditions of arid and semiarid regions contribute to dust emissions. Indeed, dust is an important indicator for assessing the degree of desertification.

Under the conditions of excessive use of land resources, global warming, and lack of water, dust storms have seriously affected socioeconomic development and environmental integrity. As a result, dust is studied as an important issue in various sciences [2,3]. Dust storms cause serious damage, including decreased visibility; road accidents; deaths of living creatures; and loss of agricultural lands, pastures, and crops. Dust storms affect the environment on a local, regional, and global scale, and ultimately affect climate directly and indirectly [4,5]. The effects on agriculture include soil degradation and air pollution, which damage vegetation by impairing the vital processes of plants, reducing production in the impacted region.

The greatest consequence of dust storms is related to the reduction of horizontal visibility. Visibility can be reduced to several meters, which causes road accidents. The World Meteorological Organization categorizes dust storms based on horizontal visibility: a weak dust storm has horizontal visibility of over 10 km, a moderate dust storm has visibility of 1 to 10 km, a severe dust storm has visibility of 200 to 1000 m, and a very severe dust storm has visibility less than 200 m indirectly [6].

A dust storm is a random phenomenon in which several variables play a role, the most important of which are parameters that relate to the near-surface meteorology and the upper atmosphere indirectly [7]. The intensity and frequency of dust storms are dependent on these parameters. Studies have shown that three major conditions are needed to create a dust incident: high wind speeds, areas with loose soil (sandy areas), and unstable atmospheric conditions [8].

The frequency of dust storm events is an indicator of environmental change and is used to measure desertification processes, changes in coverage, and the impact of human activities in arid and semiarid regions [9–11]. In recent years, as the study of natural disasters such as dust storms has increased, researchers have focused on accurately assessing the return period of incidents as a way to minimize economic losses and improve risk management.

The return period is the average time interval between occurrences of a recurring phenomenon, and is used to describe the severity and frequency of natural disasters [7]. In particular, the return period is widely used in the assessment of hydrological and meteorological natural disaster risk, as well as in design, planning, and project management [7,12]. and in creating effective early warning systems and carrying out necessary control measures based on the mechanisms of dust storm formation and data monitoring [13].

Dust storms cause irreparable damage to human health and economic sectors in Iran [14]. For instance, [15] reported that the number of respiratory patients as a consequence increasing PM10 concentration of middle east dust storm increased in Ilam, Iran during a period of one year (2015–2016). Other studies also revealed that the increase of air pollution resulting from dust storm events has led to an increase in the number of respiratory patients throughout the world, such as in Japan [16], Taiwan [17,18], Africa [19], and Cyprus [20]. Therefore, effective methods for analyzing the return period of severe dust storms can be used as guidance to manage this phenomenon in the affected regions. Due to the multivariate nature of dust storms, the use of univariate methods leads to errors in the calculation of the return period [7,21,22]. Therefore, simultaneous analysis of effective parameters in the formation of a dust storm for determining the return period is necessary and important in assessing the potential hazards of dust storm occurrence.

Given that the effective parameters in creating a dust storm event may have different marginal distribution functions, the most suitable tool for modeling the joint behavior of these variables is the multivariate copula. Copula theory is widely used for joint modeling [23–28]. There are several copula families that are used in multivariate analysis, such as Archimedean, Elliptical, and Gaussian mixture. Archimedean and elliptical copulas are widely used for joint analysis of natural disasters [7,29,30].

The advantage of copulas is the lack of limitation of the type of marginal function. The determination of the return period using copula functions can increase the accuracy of estimation compared to other functions [29,31].

In this study, bivariate analysis of severe dust storms is used to model the dust storm return period. The potential predictors considered include meteorological parameters from two spatial levels:

upper atmosphere (the free troposphere) and near surface level. Interactions between the upper atmosphere and near surface levels are the main reason for the dust storm. Therefore, the parameters of different atmospheric levels can play an important role in dust storms. For this purpose, the near surface maximum wind speed, geopotential height, and vertical velocity at different atmospheric levels (500, 850, and 1000 hPa) were selected as possible predictors of the return period of dust storms. Wind is a dynamic factor and a necessary condition for the creation and movement of dust from the surface of the earth. Many studies have considered wind speed as one of the most important parameters in the creation of dust storms [26,27,32,33]. Geopotential height indicates the atmospheric circulation pattern at upper atmosphere. In order to create dust storms, a low-pressure air mass is required. But if this low-pressure system is coordinated with the low geopotential height at intermediate atmospheric levels, and the thermodynamic relationship is established, the most suitable conditions for creating dust storms will arise. Additionally, the instability caused by the vertical velocity associated with a low pressure system can enhance conditions of the dust storm.

Most studies of dust storms have been performed using univariate methods. The hypothesis pursued here is that since dust storm occurrence and severity depends on several variables, such as wind speed, horizontal visibility, dust storm duration, and vertical velocity parameters at different levels, a multivariate approach would be useful. Most studies in the field of multivariate analysis of natural disasters are about hydrological and meteorological events such as floods and droughts [25,34–38]. Few multivariate studies have been conducted on dust storms [7,29,30], although more researchers have recently discovered the importance of multivariate analysis of dust storms. [31] investigated the factors affecting storm dust by using correlation analysis. [29] calculated the return period of 79 severe dust storms in China over 1990–2008 using bivariate Archimedean copula functions. In that study, wind speed and dust storm duration were considered as the effective parameters of dust storm. The results of the study showed that the bivariate approach modeled the dust storm return period better than a univariate one. In [7], the parameters used for dust storm multivariate analysis include the 500-hPa atmospheric longitudinal circulation index, maximum wind speed in the 10-m high near ground, and surface soil moisture. The results showed that the bivariate Frank copula was suitable for return period of less than 10 years and trivariate Frank copula was suitable to estimate return period of more than 10 years. [30] investigated the dual effect of plant growth season and severe winds on spring dust storm in China using copulas. In this study, two pairs of variables of maximum wind speed–length of dust storm and growing season start date–number of intense wind events were used for bivariate analysis. The results of this study showed that the return period on the basis of the second pair of variables was closer to reality. In the current study area of Iran, a study of bivariate return period of dust storm has been carried out by [39]. There, the dust storm return period was calculated based on wind speed and geopotential height using copula theory for Yazd province in the period of 1982–2014.

Therefore, treating dust storm occurrence as a random variable, the bivariate return period of dust storm events will be estimated on the basis of the pairs of maximum wind speed–geopotential height and maximum wind speed–vertical velocity for the period of 1982–2017 in Yazd province, central Iran. For this purpose, the copula theory will be used for joint bivariate modeling of dust storm occurrence as a function of the above predictors. This theory will model the structure of the correlation between variables to provide bivariate predictions of dust storm return period. Based on the return period of severe dust storms, an early warning system can be created.

## 2. Materials and Methods

Located in the middle of the Loot desert and the Central desert of Iran, Yazd province extends from 29 degrees 52 min to 33 degrees and 27 min north latitude and 52 degrees 55 min to 56 degrees and 37 min east longitude and covers an approximate area of 131,575 square kilometers with the population of 990,818 people (Figure 1). Due to low precipitation, high temperature, and geographical location of the province, about half of its area is covered by desert lands, which are always exposed

to wind erosion and dust storms, and it is considered one of the critical hotspots of wind erosion in Iran [40]. Dust storms occasionally cause darkening air in the area due to their high intensity. In this region, severe dust storms cause irreparable damage each year, especially in the agricultural sector. Indeed, the farmers suffer from dust events more than others. Additionally, road accidents due to decreasing vertical visibility; air pollution; and the destruction of the buildings, roads, and channels are other effects of the dust storm that occur in this area every year [41]. In addition to sand dunes, there are numerous deserts or salty plains in the province, including the Siah Kooh desert in the north, the Daranjir Desert in the east, Abarkuh Desert in the southwest, and Saghand Desert in the northeast. Meteorological data and statistics indicate that the frequency of dust phenomena, including dust storms, is high in the study area [39]. Hence, recognizing the phenomenon of dust and the return period of severe storms in this region is important for combating its harmful effects. To this end, in order to estimate the return periods of dust storms, the stormy days were extracted based on the definition of the Meteorological Organization during the statistical period from 1982 to 2017 [42]. Forty dust storm events were identified during this period. Maximum wind speed variables at 10 m above ground level; geopotential height; and vertical velocity at three levels of 500, 850, and 1000 hPa were extracted, respectively, from synoptic station data and NCEP (National Centers for Environmental Prediction) data for analysis.

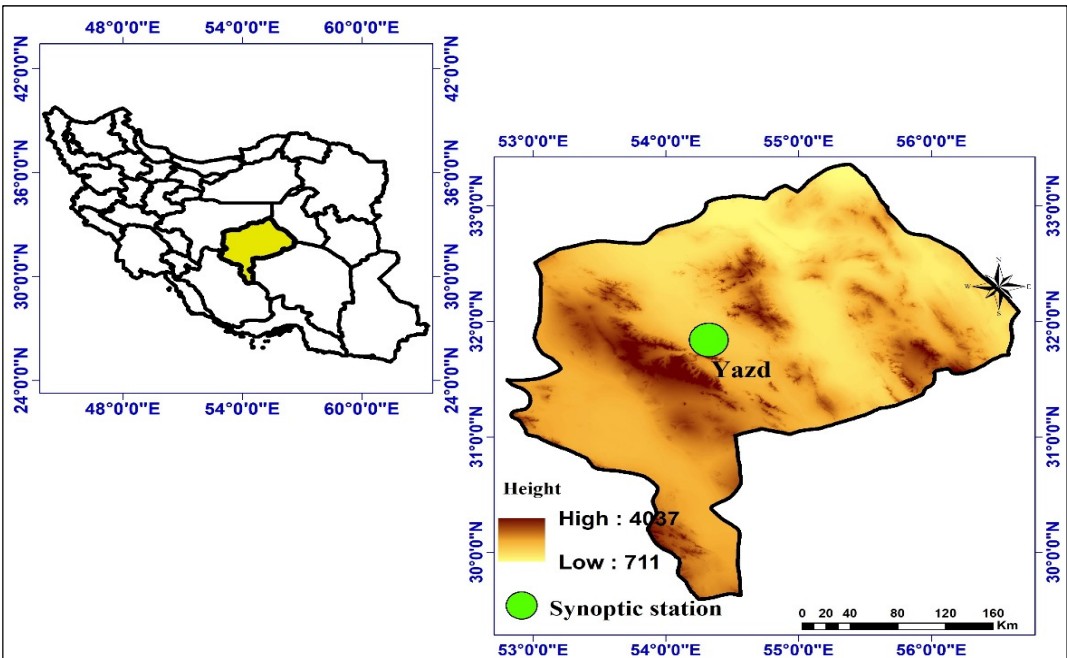

**Figure 1.** Location map of study region and synoptic station.

## 2.1. Copula Theory

To model multivariate probabilities, [43] proposed the copula function. The copula function makes it possible to combine univariate distributions with different families to create a bivariate or multivariate distribution, taking into account the dependence between variables. In other words, the copula function ($C(F_{X1}(x1), F_{X2}(x2), \ldots, F_{XN}(x_N))$) is a connection function for the association of random variables $X_1, X_2, \ldots, X_N$ with marginal functions $F_{X_1}(x_1), F_{X_2}(x_2), \ldots, F_{X_N}(x_N)$ that and the parameter of $\theta$ is defined in Equation (1) [44].

$$F(x_1, x_2, \ldots, x_N) = C_\theta \Big[ F_{X_1}(x_1), F_{X_2}(x_2), \ldots, F_{X_N}(x_N) \Big] \tag{1}$$

The most important advantage of the copula functions is the lack of limitations on the form of the marginal distributions, so that these functions can describe nonlinear and asymmetric dependence on

these variables [30], whereas other types of probabilistic distributions are modeled by the assumption that the marginal distribution structure of all the variables is identical, which can cause errors. To create a realistic copula function for multiple variables, the correlation between the variables should be known. In this research, Kendall's correlation coefficient was used to estimate the correlation within the pairs of maximum wind speed–geopotential height and maximum wind speed–vertical velocity. The value of the correlation coefficient partly determines the type of copula function. Accordingly, if the correlation between variables is positive, different types of Archimedean (Frank, Clayton, and Gumbel), elliptical (t, Gaussian, and Normal), and other types of Copula function families can be used. For negative correlation values, a smaller number of copulas can be used for modeling (Table 1).

**Table 1.** Copula functions used to bivariate analysis of dust storm.

| | | Joint CDF | Generator Function |
|---|---|---|---|
| **Archimedean Family** | Frank | $C(u,v;\theta) = \frac{1}{\theta}ln\left[1 + \frac{(e^{-\theta u}-1)(e^{-\theta v}-1)}{e^{-\theta}-1}\right], \theta \neq 0$ | $-ln\frac{e^{-\theta v}-1}{e^{-\theta}-1}$ |
| | Gumbel | $C(u,v;\theta) = exp\left\{-\left[(Lnu)^{\theta} + (-Lnv)^{\theta}\right]^{\frac{1}{\theta}}\right\}, \theta \geq 1$ | $(-lnv)^{\theta}$ |
| | Clayton | $C(u,v;\theta) = (u^{-\theta} + v^{-\theta} - 1)^{-1/\theta}, \theta > 0$ | $\frac{v^{-\theta}-1}{\theta}$ |
| | Rotated Joe | $1 - [1 - \prod_{i=1}^{m}(1 - (1-u_i)^{\theta})]^{1/\theta}$ | $-ln[1 - (1-t)^{\theta}]$ |
| | Rotated Gumbel | $C(u,v;\theta) = u + v - 1 + C(1-u, 1-v)$ | $(-lnv)^{\theta}$ |
| | Rotated Clayton | $C(u,v;\theta) = u + v - 1 + C(1-u, 1-v)$ | $\frac{v^{-\theta}-1}{\theta}$ |
| **Elliptical Family** | Student-t | $\int_{-\infty}^{t_v^{-1}(u)} \int_{-\infty}^{t_v^{-1}(v)} \frac{1}{2\pi\sqrt{(1-r^2)}}\left\{1 + \frac{x^2-2rxy+y^2}{v(1-r^2)}\right\}dxdy$ $t_v(x) = \int_{-\infty}^{x} \frac{\Gamma((v+1)/2)}{\sqrt{\pi v}\Gamma(v/2)}(1+y^2)^{-(v+1)/2}dy \quad v \neq 0$ | - |
| | Gaussian | $C(u,v) = \int_0^u \Phi\left(\frac{\Phi^{-1}(v)-\rho xy\Phi^{-1}(t)}{\sqrt{1-\rho^2 xy}}\right)dt$ | - |

### 2.1.1. Estimation of the Parameters of the Copula Functions

In order to estimate the parameters of the copula functions, nonparametric and parametric methods may be used. One nonparametric estimation approach is to use the relationship between the generator function of each copula and the Kendall correlation coefficient (Equation (2)) [45]. In the parametric method, the copula parameter is estimated by maximizing a logarithmic likelihood function (Equation (3)), where $c_{\theta}$ is copula density function, F is marginal distribution function, and $x_{1k}, x_{2k}, \ldots,$ $x_{pk}$ ($k = 1, \ldots, n$) are random variables.

$$\tau(X, Y) = 1 + 4\int_0^1 \frac{\varphi(v)}{\varphi'(v)}dv \tag{2}$$

$$L(\theta) = \sum_{k=1}^n log[c_{\theta}\{F_1(x_{1k}), \ldots, F_p(x_{pk})\}] \tag{3}$$

where $\tau$ is Kendall correlation coefficient and $L$ is a likelihood function that can be maximized.

### 2.1.2. Selecting the Copula Function

In order to select the most suitable copula function to fit the dust storm variables, the joint empirical probability values (JEPV) of the pair of variables were calculated by the empirical copula (Equation (4)) and then were compared with the values obtained from the theoretical copulas (Archimedes and elliptical families). Ordinary least squares method (*OLS*) was used to compare empirical copulas with each of theoretical copula functions. The OLS method, based on the square of the difference between

the empirical and theoretical values, can be used to choose the best function form and parameter values (Equation (5)). Two information criteria—(AIC) and (BIC)—were also used (Equations (6) and (7)) [46,47]. In Equations (4)–(7), u and v are the empirical probability values of the maximum wind speed and the geopotential height/vertical velocity, respectively; $P_{ei}$ is the empirical copula value; $P_i$ is the theoretical copula value; $k$ is the number of model parameters; $n$ is the number of observations; and $L$ is the maximum log- likelihood function.

$$C_n(u,v) = \frac{1}{n}\sum_{t=1}^{n} 1(U_t < u, V_t < v) \tag{4}$$

$$S_{OLS} = \sqrt{\frac{1}{n}\sum_{i=1}^{n}(P_{ei} - P_i)^2} \tag{5}$$

$$AIC = 2k - 2ln(L) \tag{6}$$

$$BIC = 2nLogL + kLog(n) \tag{7}$$

### 2.1.3. Analysis of Bivariate Dust Storm Return Period

The bivariate return period is based on the mean interval time between dust storm events ($E(L)$) and the joint cumulative distribution function ($C(U, V)$). Based on the definition of the return period, if N is the length of the study period, n is the number of events, L is the interval time between events, and $E(L)$ is the mean interval time of an event ($E(L)$ = N/n). In this way, the bivariate return period of a dust storm is obtained by the primary (($T_{UV}^{AND}$)) and secondary ($T_{UV}^{OR}$) modes according to Equations (8) and (9). Accordingly, the return period $T_{UV}^{AND}$ when two variables of maximum wind speed and geopotential height/vertical velocity are considered simultaneously, the bivariate return period of dust storm is based on the maximum wind speed that exceeds a certain value and the geopotential height/vertical velocity that exceeds a certain value (U >= u and V >= v) is determined, and the return period $T_{UV}^{OR}$ when the maximum wind speed exceeds a certain value or the geopotential height/vertical velocity exceeds a certain value (U >= u or V >= v) is determined.

$$T_{UV}^{AND} = T\left(U \geq u \quad and \quad V \geq v\right) = \frac{E(L)}{P(U \geq u, V \geq v)} = \frac{E(L)}{1 - F_U(u) - F_V(v) + C(F_U(u), F_V(v))} \tag{8}$$

$$T_{UV}^{OR} = T\left(U \geq u \quad or \quad V \geq v\right) = \frac{E(L)}{P\left(U \geq u \quad or \quad V \geq v\right)} = \frac{E(L)}{1 - C(F_U(u), F_V(v))} \tag{9}$$

Additionally, the univariate return period is calculated for the comparison with bivariate values based on Equation (10). In this regard, $F_X(x)$ is the marginal function of the one of the variables of maximum wind speed, geopotential height, and vertical velocity.

$$T_X = \frac{E(L)}{1 - F_X(x)} \qquad F_X(x) = Pr(X \geq x) \tag{10}$$

## 3. Results

### 3.1. Determine the Marginal Functions Of' Dust Storm Variables

It is necessary to use the copula functions for correlated predictor variables. For this purpose, the correlation between the pairs of maximum wind speed–geopotential height and maximum wind speed–vertical velocity at the three levels of 500, 850, and 1000 hPa was determined by the Kendall coefficient on the day of the dust storm and one day before the dust storm. The Kendall coefficient showed that on dust storm days, maximum wind speed as the main factor has a negative and significant relation with the geopotential height at 500 hPa and has a positive and significant relation with the

vertical velocity at 1000 hPa at 5% significant level ($p$ value < 0.05). There is no significant correlation between variables on the day before the dust storm. Table 2 shows the Kendall correlation coefficients within pair of dust storm predictor variables.

**Table 2.** Kendall correlation coefficient between two pairs of maximum wind speed–geopotential height and maximum wind speed–vertical velocity at different levels.

| Variables | 500 hPa | 850 hPa | 1000 hPa |
|---|---|---|---|
| Maximum wind speed–geopotential height | −0.29 | −0.26 | −0.19 |
| Maximum wind speed–vertical velocity | 0.14 | 0.27 | 0.3 |

In order to determine the marginal functions of the pair of variables, more than 20 univariate distribution functions were fitted to the variables, and the most appropriate functions based on Kolmogorov–Smirnov, Anderson Darling, and chi square tests were selected. Based on the results, the Wakeby distribution function was identified as the most suitable fit for maximum wind speed data and generalized extreme value function (GEV) as the most suitable fit to the geopotential height of 500 hPa and the vertical velocity at 1000 hPa (Table 3).

**Table 3.** Marginal distribution functions of dust storm variables.

| Variables | | CDF | Parameters |
|---|---|---|---|
| Maximum wind speed | Wakeby | $F(x) = \xi + \frac{\alpha}{\beta}\left(1 - \left(1 - u^{\beta}\right)\right)\frac{\gamma}{\delta}\left(1 - \left(1 - u\right)^{-\delta}\right)$ | $\alpha = 80.89$, $\beta = 12.59$, $\gamma = 4.56$, $\delta = 0.057$, $\xi = 0.46$ |
| Geopotential height 500 hPa | GEV | $F(x) = exp\left(-\left(1 + kz\right)^{-1/k}\right)$ <br> $z = \frac{x-\mu}{\sigma}$ | $\kappa = -0.54$ $\sigma = 98.19$ <br> $\mu = 5761.7$ |
| Vertical velocity | GEV | $F(x) = exp\left(-\left(1 + kz\right)^{-1/k}\right)$ <br> $z = \frac{x-\mu}{\sigma}$ | $\kappa = -0.23$ $\sigma = 0.082$ <br> $\mu = -0.038$ |

### 3.2. Choosing the Best Copula Function for Bivariate Modeling of Dust Storms

In this study, the negative correlation between the pair of maximum wind speed–geopotential height resulted in candidate copulas for bivariate modeling of the dust storm that included Frank, Gaussian, Rotated Clayton, Rotated Gumbel, Student t, and Joe. Similarly, based on the positive correlation between the pair of variables maximum wind speed–vertical velocity, the copula types considered were the Frank, Gumbel, Clayton, Gaussian, and Student t (Table 1). The parameters of the copula functions were estimated by both parametric and nonparametric methods. The Student t copula parameters, of which there are two parameters, are estimated only by the parametric method. The nonparametric method, which is based on the relationship between the Kendall coefficient (t) and the generator function of each copula, applies only to functions with one parameter. In Tables 4 and 5, the values of the copula parameters are presented based on both parametric and nonparametric methods for the pair of dust storm variable. After estimating the parameters, all selected copula functions were fitted to a pair of dust storm variables and the best fit was obtained by comparing the values of the empirical copula function against each of the above copulas as well as the OLS, AIC, and BIC criteria for the bivariate modeling of dust storms. Based on the selection criteria, Student t and Gaussian copulas were selected for bivariate modeling because of having the lowest values of AIC, BIC, and OLS (Tables 4 and 5). Figure 2 shows the q-q plot of the Gaussian copula and Student t copula for the pair of dust storm variables. The high correlation between the empirical and theoretical copulas in each pair of variables expresses the appropriateness of the Student t and Gaussian functions for the bivariate modeling of dust storm. Figure 3 shows the probability density values of the selected copula functions (Student t and Gaussian) for each of variables. The joint probability values (Figure 3) for the dust storm variables, maximum wind speed, and geopotential height have a symmetric correlation in

upper and lower tails (Figure 3a), while the maximum wind speed and the vertical velocity are not correlated in the upper and lower tails (Figure 3b).

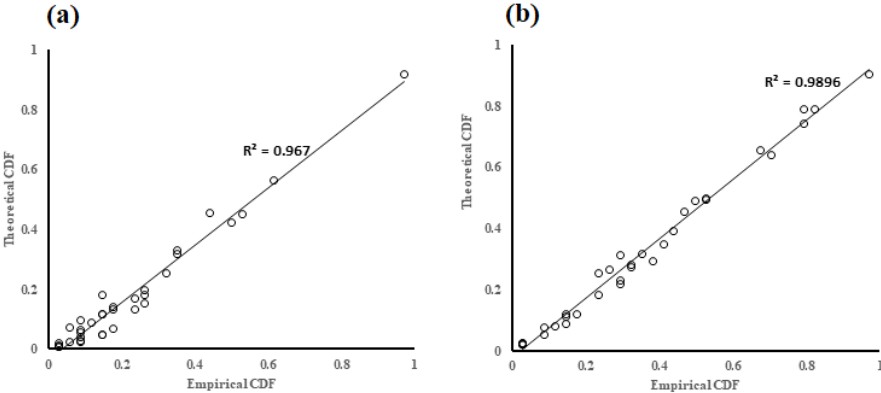

**Figure 2.** Q-Q plot for (**a**) the fitted Student t copula and (**b**) for the fitted Gaussian copula.

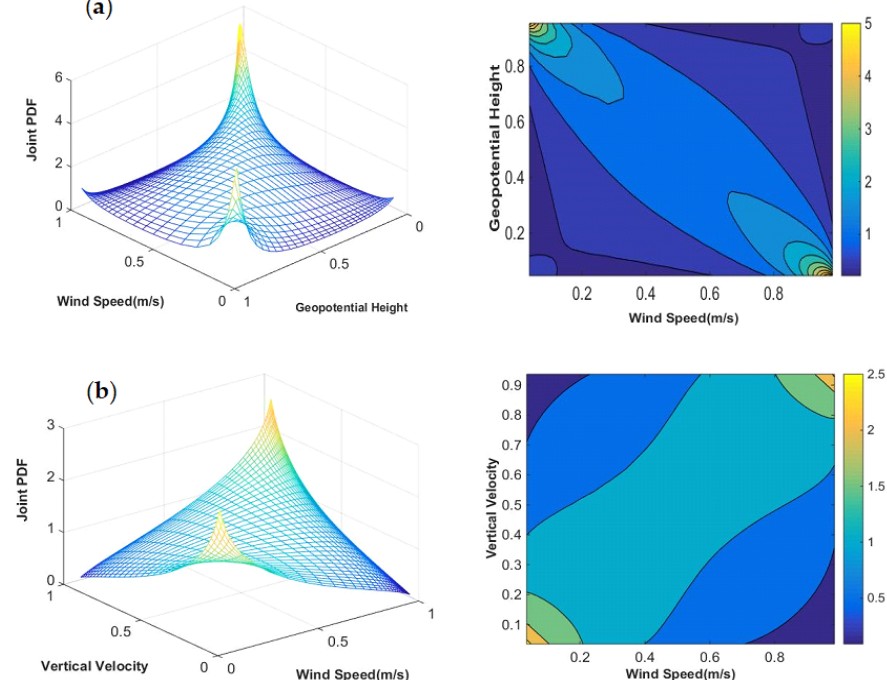

**Figure 3.** The joint probability distribution function (JPDF) and corresponding contour for each pair of dust storm variables (**a**) Student t copula, (**b**) Gaussian copula.

**Table 4.** Selection criteria of the best fit of copula functions for maximum wind speed–geopotential height.

| Copula CDF | Parametric Estimation | | | | Nonparametric Estimation | | | |
|---|---|---|---|---|---|---|---|---|
| | AIC | BIC | $S_{OLS}$ | Parameter | AIC | BIC | $S_{OLS}$ | Parameter |
| Frank | −5.38 | −3.85 | 0.3557 | −2.46 | −5.34 | −2.1 | 0.3557 | −2.3 |
| Gaussian | −2.76 | −0.53 | 0.2533 | −0.46 | −2.31 | −0.88 | 0.3533 | −0.28 |
| Rotated Clayton | −2.66 | −0.34 | 0.256 | −0.77 | −2.02 | −0.201 | 0.3590 | −0.48 |
| Rotated Gumbel | −5.5 | −4.04 | 0.354 | −2.44 | −5.13 | −2.81 | 0.3543 | −1.14 |
| Student t | −7.4 | −4.45 | 0.2312 | −0.57 2 | - | - | - | - |
| Rotated Joe | −5.15 | −3.7 | 0.3451 | −2.68 | −4.2 | −2.6 | 0.3493 | −1.47 |

**Table 5.** Selection criteria of the best fit of copula functions for Maximum wind speed–vertical velocity.

| Copula CDF | Parametric Estimation | | | | Nonparametric Estimation | | | |
|---|---|---|---|---|---|---|---|---|
| | AIC | BIC | $S_{OLS}$ | Parameter | AIC | BIC | $S_{OLS}$ | Parameter |
| Frank | −2.82 | −0.489 | 0.036 | 1.85 | −2.24 | −0.0212 | 0.046 | 2.46 |
| Clayton | −2.89 | −0.536 | 0.035 | 0.64 | −2.84 | −0.0157 | 0.047 | 0.643 |
| Gumbel | −3.1 | −0.637 | 0.031 | 1.38 | −1.92 | −0.373 | 0.045 | 1.87 |
| Gaussian | −3.81 | −2.31 | 0.029 | 0.289 | −2.28 | −1.86 | 0.042 | 0.571 |
| Student t | −2.01 | −1.98 | 0.041 | 0.87, 1.58 | - | - | - | - |

### 3.3. Joint and Conditional Probability of Dust Storm

Using the fitted copula function, the probability of dust storm can be determined based on its effective parameters. Here, the joint probability is the probability that the parameters affecting the dust storm will exceed a certain value ($P(U \geq u, V \geq v)$). Knowing of this possibility could be helpful in assessing the risk of severe dust storm and creating a dust storm warning system. The joint probability of dust storm is defined by the following relation based on the copula theory [34]).

$$P(U \geq u, V \geq v) = 1 - F_U(u) - F_V(v) + C(F_U(u), F_V(v)) \tag{11}$$

For example, for a dust storm event based on the pair of maximum wind speed–geopotential height, when wind speed and geopotential heights exceed 13 m/s and 5681 m, the probability of dust storm occurrence is 0.22. Similarly, based on the pair of variable, when the values exceed 13 m/s and −0.12, respectively, the wind speed–vertical velocity occurrence probability is 0.021 (Figure 4). Additionally, the conditional probability of the dust storms by the copula function was computed based on the threshold velocities of 6.5 m/s in the study area [40] (Equation (12)). Accordingly, the probability of dust storm occurrence by dividing the wind speed, according to the threshold velocities set at 3 levels (3, 5.6 and 13 m/s), was obtained conditionally (Figure 5). As the results showed, the probability values of the dust storm are different for the threshold levels of wind speed. Based on the wind speed–geopotential height, with increasing wind speed, the probability of dust storm occurrence will increase, while based on the wind speed–vertical velocity, the probability will decrease at higher wind speed. For example, when the wind speed is higher than 3 m/s (less than the threshold level) and the geopotential height and vertical velocity are less than a certain value (probability of occurrence equal to 0.2, $F(v) = 0.2$), the probability of dust storm occurrence are, respectively, 0.20 and 0.19. If the wind speed is higher than 6.5 m/s (threshold velocity) and the geopotential height and vertical velocity are less than a certain amount, the probability is 0.21 and 0.17, respectively. Additionally, if the wind speed is higher than the threshold speed in the study area (above 13 m/s) and the geopotential height and vertical velocity are less than the specified value, the probability is 0.37 and 0.90, respectively (Figure 5).

$$P(V \leq v | U \geq u') = \frac{F_V(v) - F_{V,U}(u', v)}{1 - F_U(u')} = \frac{F_V(v) - C(F_U(u'), F_V(v))}{1 - F_U(u')} \tag{12}$$

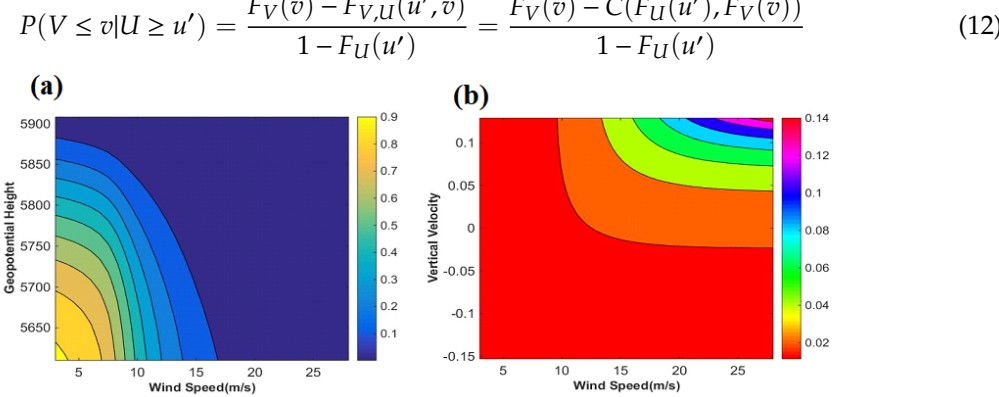

**Figure 4.** The joint probability $P(U \geq u, V \geq v)$ for (**a**) geopotential height and wind speed, (**b**): vertical velocity and wind speed, based on copula functions (left: Student t copula and right: Gaussian copula).

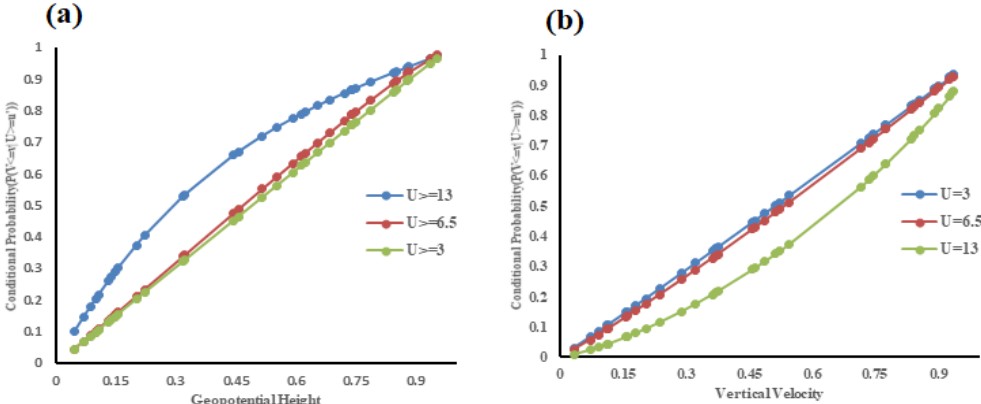

**Figure 5.** The conditional probability of geopotential height (**a**) and vertical velocity (**b**) given wind speed exceeding a certain value, $u'$.

### 3.4. Bivariate Return Period of Dust Storm

According to Equations (8) and (9), the joint return period of dust storm was calculated by fitting the Student t and Gaussian copula functions to the pair of variables. The dust storm return period, as defined, is a function of the interval time between dust storms in a study period, which is determined by the mean of these intervals. The results of calculating the return period $T_{UV}^{AND}$ and $T_{UV}^{AND}$ for the pair of variables are shown in Figure 6. The $T_{UV}^{AND}$ return period of most of the dust storm events based on the maximum wind speed–geopotential height in the study period is less than 1 year (Figure 7a) and based on the maximum wind speed–vertical velocity is less than 0.5 year (Figure 7b) Additionally, the $T_{UV}^{OR}$ return period of most events based on the pair of dust storm variables (maximum wind speed–geopotential height and maximum wind speed–vertical velocity) is less than 1 year (Figure 7c,d). The $T_{UV}^{AND}$ maximum return period was based on the maximum wind speed–geopotential height for a severe dust storm event with a horizontal visibility of 500 m and a wind speed of 29 m/s on 20 May 2016, while the $T_{UV}^{AND}$ maximum return period on the basis of the maximum wind speed–vertical velocity was on 20 February 2003, which was a less severe dust storm, with horizontal visibility of 1000 m and a wind speed of 22 m/s. The values of the univariate return period based on the maximum wind speed showed that the dust storm event has a maximum value on the same date as the $T_{UV}^{AND}$ return period (20 May 2016). However, based on the geopotential height variable, the maximum return period for the dust storm event occurred on 19 June 2010 with a lower intensity and wind speed of 7 m/s, and for the vertical velocity, the maximum return period for the dust storm event occurred on 20 February 2003. Figure 6 shows univariate return period of the dust storm events based on each variable (maximum wind speed, geopotential height, and vertical velocity). The maximum joint return period of $T_{UV}^{OR}$ was based on the maximum wind speed–geopotential height and maximum wind speed–vertical velocity on the same dates as the return period of $T_{UV}^{AND}$. Therefore, considering that the maximum return period obtained from the maximum wind speed–geopotential height matches the intensity of the dust storm events, it can be concluded that the bivariate modeling of dust storms based on maximum wind speed and geopotential heights is more suitable than that based on the maximum wind speed and vertical velocity. In other words, the return period resulting from the variables of maximum wind speed and geopotential height at the occurrence time of a severe dust storm conforms to the maximum wind speed recorded during the study period, while the return period resulting from the variables of maximum wind speed and vertical velocity at the time of the dust storm conforms with less intensity and lower wind speed. It can be concluded that dust storm modeling based on maximum wind speed and geopotential height can better describe the situation of severe dust storms in the study region.

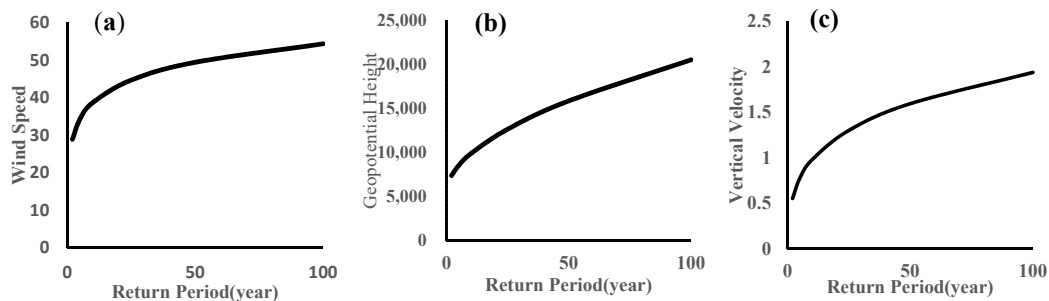

**Figure 6.** Univariate return period of dust storm events based on maximum wind speed (**a**), geopotential height (**b**), and vertical velocity (**c**).

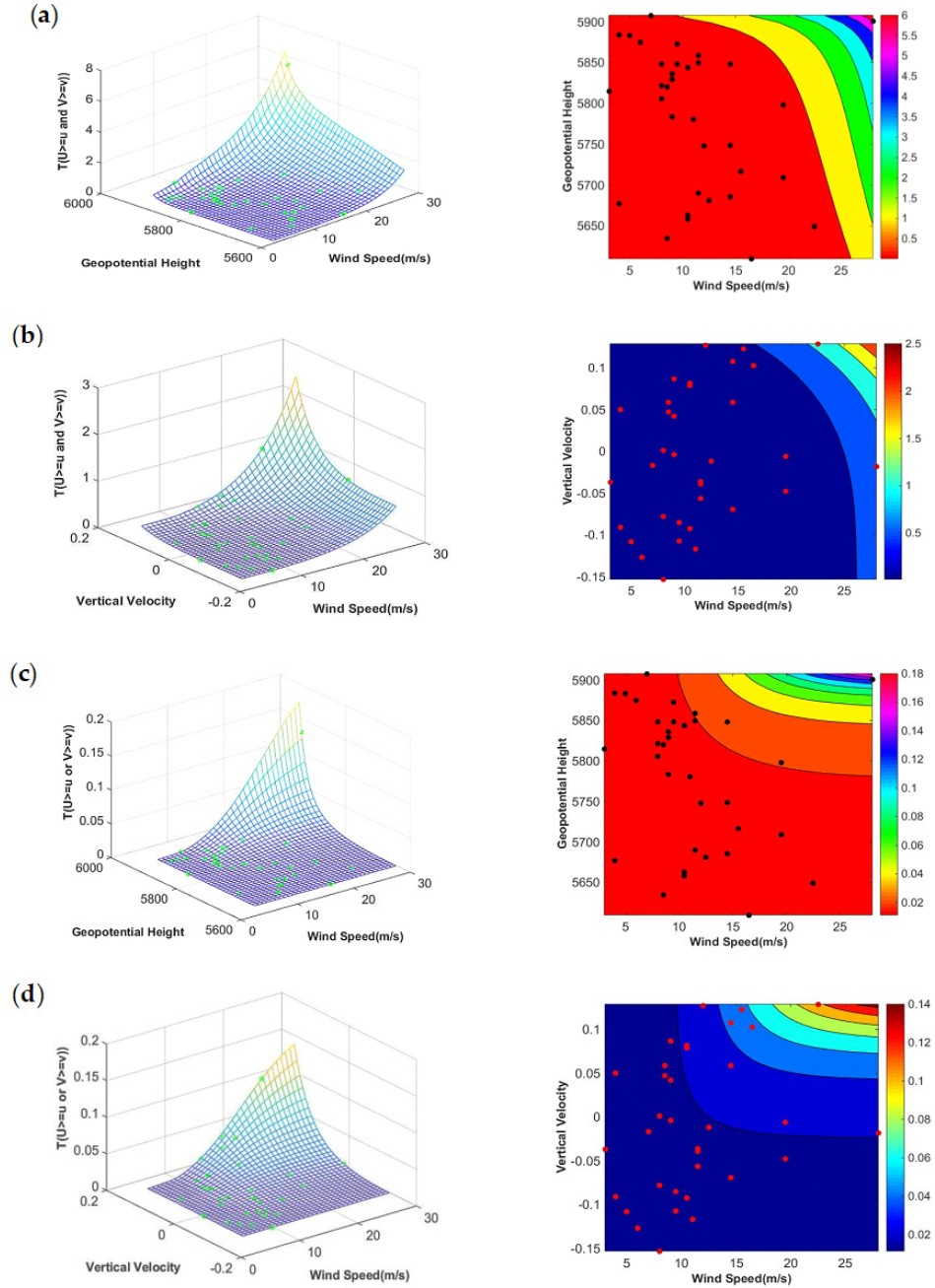

**Figure 7.** Bivariate return period (in years) and corresponding contour lines based on two pairs of dust storm variables ((**a**,**b**): $T_{UV}^{AND}$, (**c**,**d**): $T_{UV}^{OR}$).

## 4. Conclusions

Dust storms are occurring frequently as a natural disaster, partly due to global warming and desertification. Due to the repetition of this phenomenon in recent years, information about its return period is a key reference for risk assessment, as well as the forecast and warning of the medium and long-term risks [48]. The estimation of the return period of natural disasters, such as dust storm, will determine the course of the occurrence of this phenomenon in the future and allow for the assessment of its severity purposefully and quantitatively. In fact, calculating the return period can predict the occurrence time of a phenomenon on different time scales for the future.

In this study, the return period of 40 dust storm events during the statistical period was estimated using copula theory. Because the generator factors of dust storm have different probability distribution function, there is a nonlinear relationship between them. Therefore, copula theory was used to solve this problem. Bivariate modeling was performed based on the effective factors of dust storm events. These parameters were selected at two spatial levels (near-surface and upper-atmosphere levels). At near-surface level, the maximum wind speed was applied to joint analysis of severe dust storm events, while vertical velocity and geopotential height were considered as upper atmospheric level parameters.

There are few research papers that can compared with our finding in this study. However, the results of correlation analysis indicate that there is a significant negative relationship between maximum wind speed and geopotential height, and a positive and significant relationship between the maximum wind speed and vertical velocity. Based on the results of our best fit, the Student-t and Gaussian copula functions were used for bivariate modeling of storm storms. These findings were not similar to the results of [7,29], and [30], due to the different dust storm variables. Those selected Clayton and Frank copulas for bivariate modeling of dust storm.

However, our finding indicated that the bivariate return period of dust storm is closer to reality than univariate return period. These results are in agreement with [29], who showed bivariate return period of dust storm was smaller than the univariate return period.

Estimation of the occurrence time of dust storms in the future is very important to prevent the probability of hazards and for risk management. In this study, considering the random nature of dust storm, bivariate modeling was conducted to evaluate this phenomenon by copula functions.

The results of the dust storm return period showed that the values obtained from the pair of maximum wind speed–geopotential height can describe the dust storm conditions better than the pair of maximum wind speed–vertical velocity, because the occurrence time of severe dust storm and maximum wind speed, recorded at the study period, is almost similar. Generally, the finding of this study indicated that the joint modeling of dust storm events using the pair variable of maximum wind speed–vertical velocity can help provide the practical strategies to reduce the potential hazards as a consequence of severe dust storm. Finally, we propose using other copula functions such as Gaussian mixture copula to joint modeling of dust storm events. Due to nonlinear relationship between the effective parameter of dust storm event, the Gaussian mixture copula can model highly nonlinearity in the data [49].

**Author Contributions:** Conceptualization, T.M. and M.M.S.; methodology, M.M.; software, M.M.; validation, M.M., T.M. and M.M.S.; formal analysis, F.S.S.; investigation, F.S.S.; resources, F.S.S.; data curation, F.S.S.; writing—original draft preparation, M.M.; writing—review and editing, N.Y.K.; visualization, N.X.; supervision, T.M. All authors have read and agreed to the published version of the manuscript.

**Funding:** This research received no external funding.

**Conflicts of Interest:** The authors declare no conflict of interest.

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
