# Peer review of "Joint Modeling of Severe Dust Storm Events in Arid and Hyper Arid Regions Based on Copula Theory: A Case Study in the Yazd Province, Iran"

_climate, doi:10.3390/cli8050064_

Round 1

Author Response

However, this reviewer found it surprising that at least two very relevant items are missing from the references:

  1. Salvadori (2004) Bivariate return periods via 2 copulas, Statist. Methodol.,
  2. Salvadori, C. De Michele, N.T. Kottegoda and R.Rosso, Extremes in nature.

An approach using copulas, Springer, Dordrecht, 2007.

Done.

Below, a few points that need to be considered by the authors.

All the references quoted at the lines 69, 79{80, 83, 120, 134 and 155, 156 are missing from the bibliography, as are some of those listed at the lines 88, 89. The complete year is missing from (Feng et al. 201?), line 92, and (Shiau, 200?), line 286.

Done.

180 and 198: Archimedean

Done.

227: The second expression on this line is obscure; it should presumably read

FX(x) = Pr(X <=x).

Done.

449- 450 the French title of Sklar's paper is: Fonctions de repartition a n dimensions et leurs marges.

Done.

Reviewer 2 Report

line 137 occurrence instead of occurrence
line 158 do you mean the WMO? please provide a reference for the definitions
line 170 meaning of theta is not provided
line 190, 191 positions of formula expressions are too high in the line
line 192,193 what is the meaning of tau and L?
Figure 5 the unit of return period is missing

line 373 This finding was … , singular?
line 377 These results are in agreement with …
line 379 time of dust storm occurrence ?
line 388 severe instead of sever

Author Response

Line 137 occurrence instead of occurrence

Done.

Line 158 do you mean the WMO? please provide a reference for the definitions

Done.

Line 170 meaning of theta is not provided

The eta is the parameter of copula functions

Line 190, 191 positions of formula expressions are too high in the line

Done.

Line 192,193 what is the meaning of tau and L?

 is Kendall correlation coefficient and L is maximum likelihood estimation.

Figure 5 the unit of return period is missing

Done

line 373 This finding was … , singular?

Done

line 377 These results are in agreement with

Done

line 379 time of dust storm occurrence ? 

This sentence was corrected.

line 388 severe instead of sever

Done

Author Response

  • Do dust storm depends on these variables or the other way around?

A dust storm event depends on several factors which were only presented some of these parameters. However, some of the parameters such as wind speed, vertical velocity, geopotential height and air pressure directly affect the creation of a dust storm event. While, the vertical visibility and dust storm duration affect the impacts of a dust storm event.

  • What are the main difference between this research and Mirakbari et al (2018)?

In this study two pairs correlated variables, i.e., maximum wind speed- vertical velocity, and maximum wind speed- geopotential height, were considered to joint modeling of dust storm events in Yazd province, Iran. The used data set were also applied in the longer period (1982- 2017) than previous study (1982- 2014) (Mirakbari et al., 2018). So, the number of severe dust storm events (40 events) in this study is more than Mirakbari et al., (2018) (34 events).

  • Define U?

u1, u2,…un are univariate/ marginal distributions which were changed into fx1, fx2,..fxn in the manuscript.

  • Is the figure 5 a Q- Q plot?

These are the conditional probability of pair variables that were plotted using Equation. 12.

  • Figure 5 was not referenced in the text?

Done.

This manuscript is a resubmission of an earlier submission. The following is a list of the peer review reports and author responses from that submission.